# Clinical Implementation of *β-Tubulin* Gene-Based *Aspergillus* Polymerase Chain Reaction for Enhanced *Aspergillus* Diagnosis in Patients with Hematologic Diseases: A Prospective Observational Study

**DOI:** 10.3390/jof9121192

**Published:** 2023-12-13

**Authors:** Raeseok Lee, Won-Bok Kim, Sung-Yeon Cho, Dukhee Nho, Chulmin Park, In Young Yoo, Yeon-Joon Park, Dong-Gun Lee

**Affiliations:** 1Division of Infectious Diseases, Department of Internal Medicine, College of Medicine, The Catholic University of Korea, Seoul 06591, Republic of Korea; misozium03@catholic.ac.kr (R.L.); cho.sy@catholic.ac.kr (S.-Y.C.); nhodh@catholic.ac.kr (D.N.); 2Vaccine Bio Research Institute, College of Medicine, The Catholic University of Korea, Seoul 06591, Republic of Korea; skaks301@catholic.ac.kr (W.-B.K.); micropak@catholic.ac.kr (C.P.); 3Department of Biomedicine & Health Sciences, College of Medicine, The Catholic University of Korea, Seoul 06591, Republic of Korea; 4Department of Laboratory Medicine, College of Medicine, The Catholic University of Korea, Seoul 06591, Republic of Korea; yiy00@naver.com (I.Y.Y.); yjpk@catholic.ac.kr (Y.-J.P.)

**Keywords:** invasive pulmonary aspergillosis, polymerase chain reaction, hematologic neoplasms, bronchoalveolar lavage, drug resistance, fungal

## Abstract

The *β-tubulin* (*benA*) gene is a promising target for the identification of *Aspergillus* species. Assessment of the clinical implementation and performance of *benA* gene-based *Aspergillus* polymerase chain reaction (PCR) remains warranted. In this study, we assessed the analytical performance of the BenA probe PCR in comparison with the Aspergenius kit. We prospectively collected bronchoalveolar lavage (BAL) fluid via diagnostic bronchoscopy from adult patients with hematologic diseases. *BenA* gene-based multiplex real-time PCR and sequential melting temperature analysis were performed to detect the azole resistance of *Aspergillus fumigatus*. In total, 76 BAL fluids in 75 patients suspicious of invasive pulmonary aspergillosis (IPA) were collected. Before the application of PCR, the prevalence of proven and probable IPA was 32.9%. However, after implementing the *benA* gene-based PCR, 15.8% (12 out of 76) of potential IPA cases were reclassified as probable IPA. The analytical performance of the BenA probe PCR in BAL samples was comparable to that of the Aspergenius kit. The diagnostic performance was as follows: sensitivity, 52.0%; specificity, 64.7%; positive predictive value, 41.9%; negative predictive value, 73.3%; positive likelihood ratio, 1.473; and negative likelihood ratio, 0.741. Moreover, *benA* gene-based *Aspergillus* PCR discriminated all major sections of *Aspergillus*, including cryptic species such as *Aspergillus tubingensis.* Sequential melting temperature analysis successfully detected 2 isolates (15.4%) of *A. fumigatus* carrying resistant mutations. *BenA* gene-based *Aspergillus* PCR with melting temperature analysis enhances diagnostic accuracy and detects not only cryptic species but also resistant mutations of *A. fumigatus*. It shows promise for clinical applications in the diagnosis of IPA.

## 1. Introduction

Invasive pulmonary aspergillosis (IPA) has been a significant concern among patients with hematologic malignancies and is emerging as a threat for individuals with severe respiratory viral infections, including coronavirus disease 2019 and influenza [1,2,3]. Delayed treatment is recognized to increase mortality; hence, early treatment with antifungal agents is emphasized, but diagnosis of IPA remains challenging [4]. Histological examinations and cultures have limitations, making indirect microbiologic evidence crucial for the diagnosis of invasive pulmonary aspergillosis (IPA) [1,4]. Although galactomannan (GM) has been used as a non-culture method for the diagnosis of IPA, the associated low accuracy, the influence caused by antifungal treatment (AFT), and different turnaround times for each institution necessitate the discovery of a faster and more sensitive biomarker [5,6,7,8]. 

Detection of *Aspergillus* DNA through polymerase chain reaction (PCR) is a promising non-culture-based method for the diagnosis of invasive aspergillosis (IA) [9]. It has been expected to replace or supplement GM, since there is a minimal decrease in sensitivity by AFT, no false positive results caused by antibiotics such as piperacillin/tazobactam, and no effect on the metabolism in the kidney and liver, unlike in the GM assay [10,11,12,13,14]. However, it has not been implemented in most institutions due to a lack of standard techniques, lack of skilled professionals, and limitations in quality control [13]. In addition, the performance and results of *Aspergillus* PCR were diverse, owing to a heterogenous population, different samples, and methods [15,16]. Although several commercial kits are available, they are not used widely in clinical practice, especially in third- and second-world countries, due to their high cost, limitations in species identification, and false positivity due to internal transcribed spacer (ITS) region-based sequencing [17,18,19].

Although the *β-tubulin* gene (*benA*) may have lower sensitivity compared to commonly used ITS-based methods, it is effective for detailed identification of *Aspergillus* species and is used as a marker for phylogenetic analysis of filamentous *Ascomycetes* [19,20]. A recent in vitro study reported that *benA* gene-based multiplex PCR showed rapid turnaround time and superior discrimination for *Aspergillus* species, with no false positivity due to cross-reaction, which are shortcomings of ITS-based molecular identification [19]. Although it is a promising diagnostic PCR method that overcomes the limitations of ITS-based analysis, its clinical applications remain limited.

Detecting azole-resistant strains is another challenging issue owing to the increasing incidence of resistant *Aspergillus* species [21]. Detection of resistance from culture delays diagnosis and increases the mortality rate of patients with IPA [22]. Commercially available kits can detect cytochrome P450 sterol 14α-demethylase (*cyp51A*) gene mutations and tandem repeats (TR) in the promotor region; however, it is time-consuming and costly [23]. Sequencing an independent high-resolution melting assay to rapidly detect azole resistance through common mutations has been reported [24]. However, its implementation in clinical practice remains to be fully elucidated.

Therefore, this study investigated the diagnostic performance and clinical benefits of an in-house *Aspergillus* multiplex real-time PCR assay based on the *benA* gene in bronchoalveolar lavage (BAL) fluid in patients suspected of IPA. We also performed a sequential melting temperature analysis to identify TR in the promotor region of the *cyp51A* gene of *Aspergillus fumigatus* isolates to verify the process time, cost-saving potential, and possible clinical use of this approach.

## 2. Materials and Methods

### 2.1. Study Patients

This prospective observational study was performed between April 2021 and March 2022 at the Catholic Hematology Hospital. The inclusion criteria were as follows: (i) adult patients (>19 years), (ii) underlying hematologic diseases regardless of disease status, (iii) pulmonary infiltrates, and (iv) bronchoscopy for diagnostic purposes. All patients who satisfied the inclusion criteria and provided signed informed consent were included in this prospective cohort study. GM, PCR, and clinical data were prospectively collected for each episode; furthermore, one recurrent episode after the completion of IPA treatment was considered an independent episode.

Bronchoscopy was performed by pulmonologists, and routine fungal cultures were included. After the identification of pulmonary infiltration on chest computed tomography, BAL was performed in a wedge position within the selected segment [25]. In cases with a high risk of bleeding or severe hypoxia, the total volume of instilled saline was adjusted according to expert opinion. This study was approved by the Institutional Review Board of Seoul St. Mary’s Hospital (No. KC20TNSI0417) and was conducted in accordance with the Declaration of Helsinki, 2013, Good Clinical Practice. 

### 2.2. Definitions

IPA was classified as proven, probable, possible, and no IPA according to the revised version of the European Organization for Research and Treatment of Cancer Invasive Fungal Infections Cooperative Group Mycoses Study Group (EORTC/MSG) [26]. Proven cases are mainly defined by culture and identification from sterile materials. Cases of probable IPA require host factor, clinical features, and mycologic evidence such as GM or PCR [26]. For the performance evaluation of *Aspergillus* PCR, the PCR results were excluded to classify the IPA episodes. True IPA-positive cases were classified as either proven or probable [27]. However, this classification may be susceptible to bias and might present challenges in applying the results to clinical decision-making, as patients with possible IPA are often treated as IPA. To address this potential bias and broaden clinical applications, we conducted a sensitivity analysis employing different criteria for defining IPA-positive cases. The GM assay (Platelia *Aspergillus* EIA, Bio-Rad, Hercules, CA, USA) in BAL fluid and serum was performed according to the manufacturer’s instructions. GM positivity was assessed using a positivity threshold of ≥1.0 optical density index in blood, and BAL was used [26]. As for PCR, only duplicate positive results were considered positive as recommended in the consensus guidelines [26]. 

### 2.3. Evaluating the Analytic Performance of the BenA Probe Multiplex Aspergillus PCR Assay

We assessed the analytical performance with clinical samples, using BAL fluid samples that tested negative for *Aspergillus* in culture and PCR. We spiked 120 BAL samples with freshly cultured conidia of *Aspergillus fumigatus*, *Aspergillus terreus*, *Aspergillus flavus*, and *Aspergillus tubingensis* (*Aspergillus* section *Nigri*). To avoid any concentration bias, we included high (1 × 10^7^ conidia), medium (1 × 10^5^ conidia), and low (1 × 10^3^ conidia) concentrations of isolated conidia in 1 mL of BAL fluid. Subsequently, we performed DNA extraction and carried out multiplex PCR analysis using the BenA probe. Additionally, for comparative purposes, we conducted an analysis using the commercially available AsperGenius^®^ multiplex kit by PathoNostics (PathoNostics, Maastricht, The Netherlands).

### 2.4. Fungal DNA Extraction from Bronchoalveolar Lavage Fluid

A total of 76 BAL fluid samples were obtained from the patient group and were used for DNA extraction. The experiment was performed by applying the method described in a previous study [28]. Roughly, samples were divided into compartments, bead beating was performed, and DNA was extracted using a commercially available kit. The detailed DNA extraction method from BAL fluid is described in the Appendix A. Figure 1 shows the schematic diagram of the overall experimental method. To confirm whether DNA extraction was properly performed using the culture-negative BAL solution, the following experiment was performed. After adding 10-fold diluted *A. fumigatus* conidia 1 × 10^5^ to 1 mL of BAL solution and extracting *Aspergillus* DNA by using the above method, probe real-time PCR was performed as follows. After sequentially diluting the *Aspergillus* positive control (APC) samples and performing PCR in the same manner, a standard curve was drawn to confirm that DNA was properly extracted from the conidia.

### 2.5. In-House β-Tubulin Gene-Based Aspergillus PCR and Melting Temperature Analysis Method

Experiments were conducted using primers and probes designed in previous studies [19,20]. Briefly, a probe was designed with the *Aspergillus* section-specific region as a target in the *benA* gene, and a primer was selected accordingly. The primer and probe sequences used in multiplex probe real-time PCR are listed in Appendix A. In addition, SYBR Green real-time PCR was performed using primers that amplify the TR region target of the promoter region of *cyp51A*; the primers used are listed in Appendix A. For multiplex probe real-time PCR, the LightCycler^®^ 480 Probes Master Kit (Roche Diagnostics Co., Basel, Switzerland) and Roche LightCycler^®^ 480 Instrument II (Roche Co.) were used. The primer and probe concentrations used in the experiment were 0.4 μM and 0.1 μM, respectively. The amplification process was conducted as follows: 10 min of pre-denaturation at 95 °C, followed by 40 cycles of denaturation at 95 °C for 15 s, annealing at 58 °C for 15 s, and extension at 72 °C for 15 s. Three pairs of probes were used in one well and 3 pairs, as listed in Appendix A, were divided into two runs. Asco 1F9, Fumi 1R2, and Nig 1R26 probe procedures were first performed, while Flavi 1F18 and Terrei 1R29 were performed after Asco 1F9 amplification, and TR melting was performed after Fumi 1R2 amplification. Lastly, Tub_1R21 was amplified after Nig 1R26 amplification (Figure 1). In all experiments, 10-fold serial dilutions of APC developed in previous studies were used as positive controls and standard curves [19]. A schematic diagram of the two-step PCR method is shown in Figure 1.

TR melting experiments were performed using a LightCycler^®^ 480 SYBR Green I Master Kit (Roche Co.) and Roche LightCycler^®^ 480 Instrument II (Roche Co.). The experimental method was similar to that of a previous method, and annealing was performed at 58 °C for 30 s. The experimental results were automatically analyzed using the LightCycler^®^ 480 Software version 1.5.0 SP3 (Roche Co.) program, and the cutoff was selected as positive when it was within the standard curve range of APC based on the limit of detection value.

### 2.6. PCR Inhibitor Test

A SPUD inhibitor PCR test was performed using DNA extracted from BAL fluid [29], as previously described [15]. In the experiment, 10^5^ gene copies of SPUD DNA were spiked into the DNA extracted from BAL fluid, and probe real-time PCR was performed. As a control, an equal amount of SPUD DNA was added to DW. A difference in crossing point (Cp) values of the two PCRs within 1 suggests that the PCR inhibitor had no effect [29]. 

### 2.7. Study Outcomes of Interest

The primary outcomes were performance- and species-discriminative function of *benA*-based diagnostic *Aspergillus* PCR in BAL fluid. The secondary objective was to evaluate the clinical use of sequential melting temperature analysis to detect azole resistance mutations in *A. fumigatus* isolates.

### 2.8. Statistical Analysis

The medians with ranges for continuous variables and frequencies with percentages for categorical variables were presented. Possible IPA cases were classified as no IPA; however, they were grouped with proven/probable IPA cases or were excluded altogether for sensitivity analysis [30]. Diagnostic parameters, such as sensitivity, specificity, positive predictive value (PPV), negative predictive value (NPV), positive likelihood ratio (LR+), negative likelihood ratio (LR−), and diagnostic odds ratio (DOR) were calculated and presented with 95% confidence intervals (CI). Statistical analyses were performed using SAS version 9.4 (SAS Institute, Inc., Cary, NC, USA).

## 3. Results

### 3.1. Characteristics of Study Patients

A total of 76 BAL fluids from each episode in 75 patients with hematologic diseases were prospectively collected. Patients with acute leukemia and those who had undergone chemotherapy accounted for 64% (48 of 75) and 50.7% (38 of 75), respectively. Serum GM was performed in all episodes of all patients, and BAL GM was performed in all except for three episodes. Of the 76 episodes, 25 (32.9%) were from proven or probable IPA according to the revised EORTC/MSG consensus criteria. Overall, 59.2% (45 of 76) of samples were obtained from patients who had been treated with antifungal agents before bronchoscopy. The detailed patient characteristics are listed in Table 1 and Table 2.

### 3.2. Analytic Performance of the BenA Probe Multiplex Aspergillus PCR Assay

Table 3 shows the results of clinical performance analysis using the BenA probe and Aspergenius kit. First of all, in the analysis using the BenA probe, Fumi 1R2 showed an amplification efficiency of 120/120 (100%) for positive samples and 112/120 (93.3%) for negative samples. The average (±standard deviation [SD]) Cq values for positive samples were 19.255 (±0.982), 28.381 (±0.543), and 35.665 (±0.660) for each concentration. The average Cp value for false positive results was 36.049 (±1.224). Although the number of samples is limited, it can be confirmed that the Cp value is larger than that of positive samples. Analysis values of PPV 100%, NPV 93.33%, sensitivity 93.75%, and specificity 100% were obtained.

In the Aspergenius kit, *A. fumigatus* showed an amplification efficiency of 120/120 (100%) for positive samples and 116/120 (96.7%) for negative samples. In the Aspergenius kit, *A. fumigatus* showed an amplification efficiency of 120/120 (100%) for positive samples and 116/120 (96.7%) for negative samples.

Likewise, the result values of other probes can be confirmed as shown in Table 3, and the Cp values were similarly measured. However, the Cp value of the multi-amplification target probe Aspergenius kit was found to be lower than that of the single-amplification target BenA probe. On the other hand, it was confirmed that there was no difference in clinical performance (PPV, NPV, sensitivity, specificity) compared to the Aspergenius kit, and it was confirmed that clinical sample analysis was possible using this kit.

### 3.3. Fungal DNA Extraction in Bronchoalveolar Lavage Fluid

We confirmed that *Aspergillus* DNA was extracted and PCR amplified similarly to APC. Therefore, DNA extraction and *benA*-based *Aspergillus* PCR were performed from clinical specimens using this method. Figure 2 shows the results of real-time PCR using DNA extracted from serially diluted APC and BAL fluid spiked with conidia. APC serial dilution from 1 × 10^5^ gene copy number can confirm that standard curves are drawn at regular intervals. Theoretically, 1 μL of the elution buffer spiked with conidia 1 × 10^5^ is expected to contain 1 × 10^3^ DNA. As a result of spiking serially diluted conidia corresponding to 1 μL of BAL fluid and extracting DNA, it can be observed that the 1 × 10^3^ gene copy number of APC and the Cp value of the sample spiked with 1 × 10^5^ are similar. The experimental results of the steps below came out similar to the standard, thus confirming that the DNA extraction protocol was normally performed and that the BenA probe real-time PCR was normally amplified in clinical samples. Therefore, we conducted *benA Aspergillus* PCR experiments using clinical samples.

### 3.4. Diagnostic Performance of β-Tubulin Gene-Based PCR in Bronchoalveolar Lavage Fluid

*Aspergillus* PCR positivity from BAL fluid was 40.8% (31 of 76). *Aspergillus* section *Nigri* was predominant (15 of 31), followed by section *Fumigati* (13 of 31). Among *Aspergillus* section *Nigri*, *A. tubingensis* accounted for 13.3% (2/15) (Table 2 and Appendix A). Co-infection with *Aspergillus* sections *Fumigati* and *Nigri* was identified in three BAL fluids. *Aspergillus* section *Fumigati* was only amplified for *A. fumigatus*. As a result of SPUD analysis conducted to identify PCR inhibitors by BAL fluid in the extracted DNA, it was confirmed that the difference in Cp value between the DNA spiked in DW and the DNA extracted from BAL remained within 1 (Appendix A). Therefore, it was confirmed that there were no PCR inhibitors in the DNA extracted from BAL fluid. 

During the study period, 15.8% (12 of 76) of the episodes classified as possible IPA due to a lack of mycological evidence were reclassified as probable IPA according to the positive PCR result (Table 2). To calculate the diagnostic performance, only proven and probable IPA patients were included as a positive population. The diagnostic performance of *Aspergillus* PCR in BAL fluid was as follows: sensitivity, 52.0%; specificity, 64.7%; PPV, 41.9%; NPV, 73.3%; LR+, 1.473; and LR−, 0.741 (Table 3). 

### 3.5. Sensitivity Analysis

We performed three sensitivity analyses to assess differences in the results based on the classification of possible IPA cases. Regardless of the classification of possible IPA, *benA* gene-based multiplex PCR showed high specificity consistently (Table 4). 

### 3.6. Detection of Azole Resistance with the Melting Temperature Analysis Method 

Among the 13 isolates of *A. fumigatus*, 2 isolates (15.4%) showed a melting temperature of 85.6 ± 0.6 °C, indicating a TR mutation in the promoter. The melting temperature was 83.0 ± 0.3 °C in other isolates of *A. fumigatus* (11 of 13). Detailed results are presented in Figure 3.

## 4. Discussion

The *benA* gene-based *Aspergillus* real-time multiplex PCR method demonstrated comparable analytical performance to the *28S rRNA* gene-targeted method. Diagnostic performance with BAL fluid indicated high specificity and NPV. Furthermore, *benA* gene-based PCR identified all major *Aspergillus* sections (*Fumigati*, *Nigri*, *Falvi*, and *Terrei*) and discriminated emerging cryptic species, such as *A. tubingensis* in section *Nigri.* The sequential melting temperature analysis method successfully detected azole resistance based on the difference in melting curve between wild type and azole resistant *A. fumigatus* in clinical practice. 

The analytical and clinical performance of *benA* gene-based PCR for diagnostic purposes is comparable to that of the ITS region-based PCR methods [30]. Irrespective of the definition of IPA, high specificity was observed in the sensitivity analysis of *benA*-based PCR; high specificity and NPV were observed upon the grouping of possible IPA cases together with non-IPA ones. Moreover, high specificity and PPV were seen on possible IPA cases together with proven or probable IPA cases. Therefore, the *benA* gene-based PCR method might be implemented not only for the purpose of diagnosis and treatment in clinical practice characterized by its high PPV but also for stringent research criteria in IPA studies, characterized by its high NPV.

The results of this study revealed a relatively low sensitivity compared with that observed in previous studies, which could be affected by several factors. The BenA probe-based PCR is highly specific and shows no false positivity due to cross-reactivity [19]. Further, PCR positivity was defined only by duplicated results, unlike previous studies in which even single positive results were considered positive [30]. Lastly, approximately 60% of the study patients were administered anti-mold active AFT for prophylactic, empirical, or targeted purposes prior to bronchoscopy; anti-mold active AFT may lower the sensitivity of *Aspergillus* PCR [5,17]. PCR, especially when used with GM assays, has shown superior clinical efficacy due to different shedding kinetics compared to *Aspergillus* PCR alone [5,31,32]. Therefore, combining *benA gene*-based multiplex *Aspergillus* PCR with non-culture methods like GM enhances diagnostic accuracy for IPA, addressing issues related to low sensitivity.

In this study, we replaced ITS sequencing with *benA*-based multiplex real-time PCR to overcome the limitations of existing methods, which often lead to false positives and difficulty in identifying major sections and emerging cryptic species. [19,33]. *BenA*-based in-house multiplex *Aspergillus* PCR successfully identified strains not limited to *Aspergillus* section *Fumigati* but also the sections *Nigri*, *Flavi*, and *Terrei* through the standardized protocol suggested by the European *Aspergillus* PCR Initiative [31]. In addition, it distinguished emerging cryptic species that could not be identified using ITS region-based PCR.

While the primary focus of probe real-time PCR kits, such as Aspergenius, is well-established, limited research covers a broad spectrum of species with a single primer set. We have the advantage of detecting all four sections commonly observed in clinical practice with a single primer set, even capturing a wide range of fungal species belonging to *Ascomycetes*. Based on this, our approach successfully identifies all such species [19]. Additionally, the timeframe for molecular diagnostic experiments, including DNA extraction, aligns with existing studies. Compared to commercialized kits, there is no difference in performance, offering a potential advantage in reducing the unit cost of diagnosis. Although sensitivity is lower than ITS, clear identification is possible within the standard curve range, enhancing diagnostic utility due to its heightened specificity.

The increase in resistant *Aspergillus* species is another emerging public health concern along with the increase in cryptic species in IPA treatment [34,35]. The high mortality and treatment failure rates of IPA due to azole-resistant *Aspergillus* strains necessitate a faster and more accurate test of resistance [18,36]. Antifungal susceptibility tests using fungal isolates have the limitations of low culture-positive rate and long turnaround time. Although the well-known azole resistance mutation of *A. fumigatus* can be identified using commercially available kits, additional kits are required, which increases the time and cost and provides insufficient clinical validation [17].

In this study, sequential melting temperature analysis was performed to identify azole resistance through the promoter mutation of *cyp51A* gene in *A. fumigatus* isolates [19]. We detected two isolates (15.4%) of *A. fumigatus* with different melting temperatures. This is higher than the resistance rate reported in a previous study [33]. The prevalence of resistant isolates in clinical samples is higher than that of environmental isolates [37]. In a previous study, we included environmental *Aspergillus* isolates, which may explain the high resistance rate in this study; the increasing trend of azole resistance might be another possibility [38]. 

This study had several limitations. First, the study results were based on the use of only diagnostic-driven *Aspergillus* PCR through BAL fluid of patients suspected of IPA. We designed this study as a diagnostic-driven test of IPA since samples obtained from the site of infection showed higher sensitivity compared with the blood-based PCR assay due to the high burden of *Aspergillus*, and routine surveillance might increase the chance of overestimation by colonization [10,17,32,39]. Therefore, the findings of this study should not be extrapolated in the case of blood-based PCR assay or in critically ill patients without underlying hematologic diseases [26,40]. Second, we did not directly compare the performance of *benA* gene-based PCR and previously used ITS-based methods in the same BAL fluid samples. For implementing *benA* gene-based PCR in clinical practice and confirming its diagnostic superiority, further research is warranted. Lastly, although clinical utility was evaluated by performing sequential melting temperature analysis to identify the azole-resistant mutations, the clinical relevance of the treatment response depending on the presence of the mutation could not be fully elucidated due to the small number of mutant isolates. 

Despite these limitations, our study has several strengths. To the best of our knowledge, this study is the first to demonstrate clinically reliable performance and superiority of the species discriminative function of *benA* gene-based multiplex real-time PCR. The standardized in-house PCR method proposed herein could be implemented and effectively used in a clinical setting and is a more valuable method in the era of cryptic species of *Aspergillus*. Furthermore, this study displayed the efficacy of the in-house sequential melting temperature analysis method to detect azole resistance in *A. fumigatus* isolates. 

## 5. Conclusions

*BenA* gene-based multiplex real-time *Aspergillus* PCR successfully identified four major sections mainly found in *Aspergillus*, including cryptic species, that have recently been proven clinically significant. The diagnostic performance of *Aspergillus* PCR using BAL fluid is acceptable for implementation in clinical practice. Sequential melting temperature analysis is a time-saving and cost-effective method for detecting azole resistance in clinical practice.

## Figures and Tables

**Figure 1 jof-09-01192-f001:**
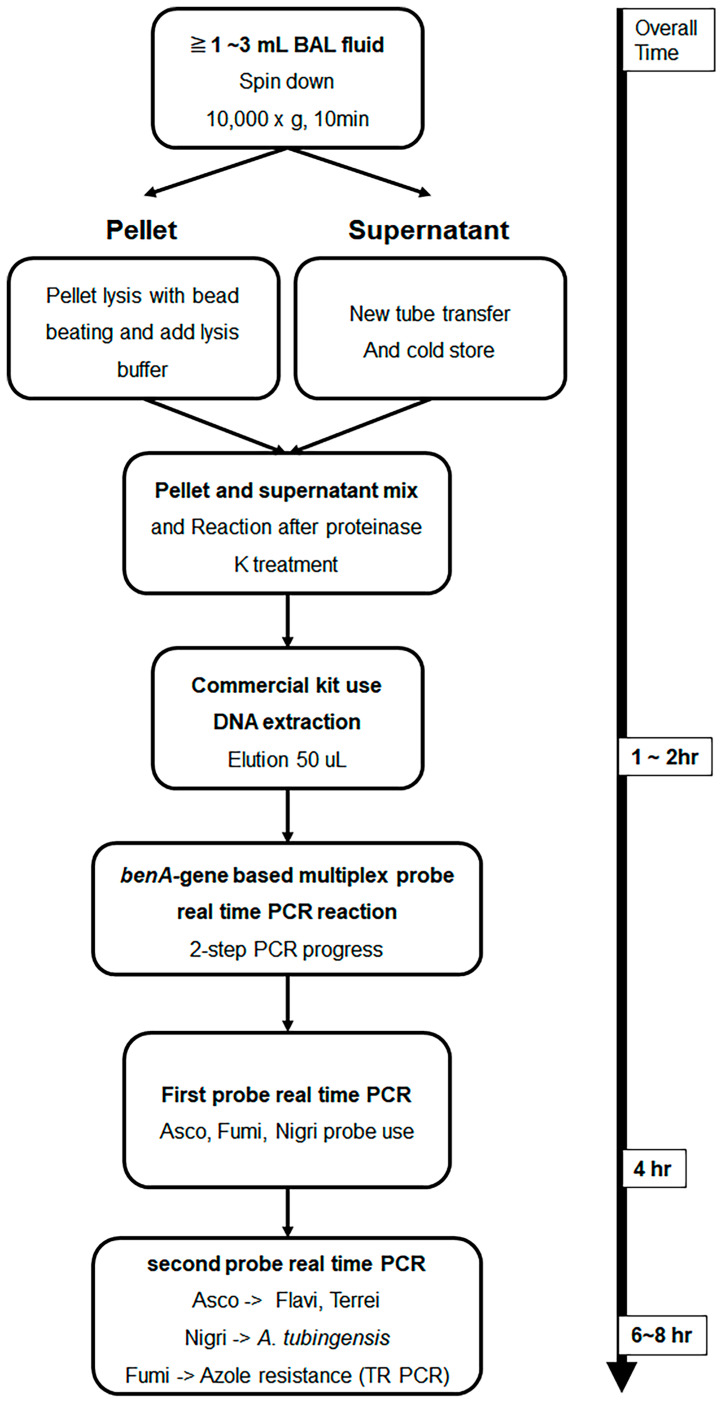
Schematic illustration of *Aspergillus* DNA extraction from bronchoalveolar lavage fluid and *β-tubulin* gene-based multiplex probe real-time *Aspergillus* PCR. BAL fluid samples were DNA extracted using the method presented in this text. Then, species analysis was performed by performing multi-probe real-time PCR, which had been studied previously, with *benA* as the target. Multiplex probe real-time PCR can be performed in one run; however, real-time PCR is designed to minimize interference between probes and distribute them more efficiently by dividing them into two rounds. Samples amplified with a section of *Fumigati* probes were subjected to SYBR Green real-time PCR using primers targeting the TR region present in the promoter of *cyp51A* to determine azole resistance. BAL, bronchoalveolar lavage; *benA*, β-tubulin; *cyp51A*, cytochrome P450 sterol 14-alpha demethylase; PCR, polymerase chain reaction; TR, tandem repeat.

**Figure 2 jof-09-01192-f002:**
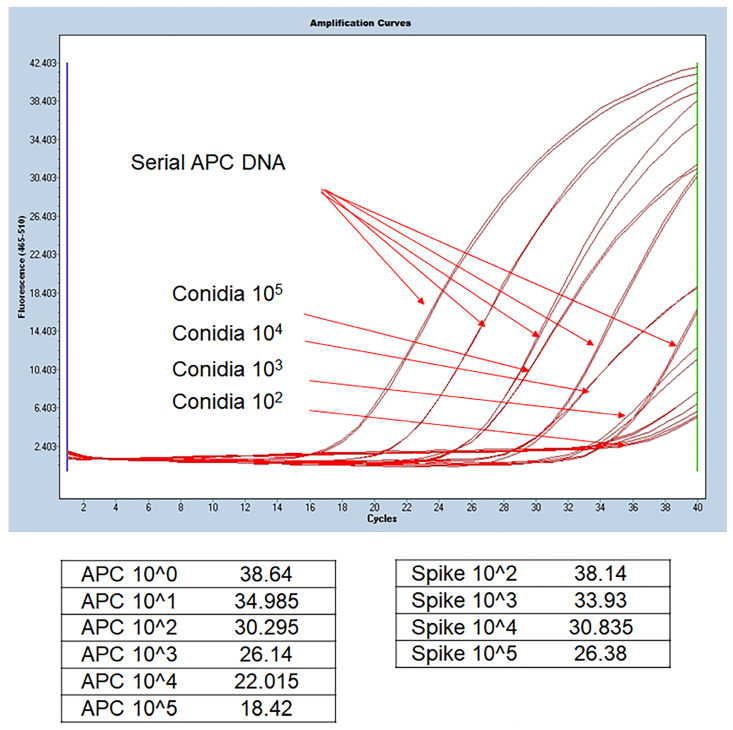
Comparison of DNA extracted from BAL fluid and *benA Aspergillus* positive control. *Aspergillus* conidia was serially spiked on negative BAL fluid samples, and the extracted DNA was used in the experiment. Quantitative values included in the extracted DNA were derived by comparing the Cp values of the extracted DNA and the serial dilution of APC DNA copy number. 1 μL was used for 100 μL of extracted DNA, and amplification similar to the Cp value of the corresponding APC concentration was confirmed. APC, *Aspergillus* positive control; BAL, bronchoalveolar lavage; Cp, crossing point.

**Figure 3 jof-09-01192-f003:**
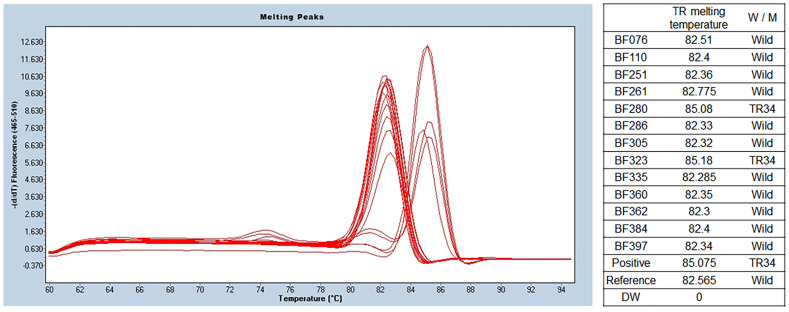
TR SYBR green PCR melting peak results for azole resistance. As a result of *benA Aspergillus* PCR, a total of 13 samples were identified as *A. fumigatus*. TR SYBR green PCR was performed using this, and different melting temperatures were confirmed in the two of them. The wild type was confirmed at 82.4 ± 0.13 °C, and two that seemed to have TR34 mutations were identified at 85.6 ± 0.6 °C. *BenA*, *β-tubulin*; TR, tandem repeat.

**Table 1 jof-09-01192-t001:** Characteristics of the study patients.

Variables	Number (%) Total = 75
Age, median	58 (48–66)
Sex, male	46 (61.3)
Hematologic diseases	
AML	27 (36.0)
ALL	21 (28.0)
MDS	17 (22.7)
Others *	10 (13.3)
Treatment status	
Induction/Re-induction	27 (36.0)
Consolidation	11 (14.7)
Undergoing HSCT	0 (0.0)
Post-HSCT state	13 (17.3)
Other chemotherapy	15 (20.0)
BSC	9 (12.0)

Data are presented as *n* (%) unless otherwise noted. ALL, acute lymphocytic leukemia; AML, acute myeloid leukemia; BSC, best supportive care; HSCT, hematopoietic stem cell transplantation; MDS, myelodysplastic Syndrome; IQR, interquartile range. * Other underlying diseases included 3 multiple myeloma, 3 lymphoma, 2 chronic myeloid leukemia, 1 severe aplastic anemia, and 1 myelofibrosis patient.

**Table 2 jof-09-01192-t002:** Diagnostic galactomannan and PCR results for each episode.

Variables	Number (%) Total = 76
IPA criteria (Before PCR applied)	
Proven	2 (2.6)
Probable	23 (30.3)
Possible	30 (39.5)
None	21 (27.6)
IPA criteria (After PCR applied)	
Proven	2 (2.6)
Probable	35 (46.1)
Possible	18 (23.7)
None	21 (27.6)
Culture positivity	3 (3.95)
Serum GM	76 (100)
Positive rate	22 (28.9)
BAL GM	73 (96.1)
Positive rate	18 (24.6)
BAL PCR	76 (100)
Positive rate, duplicate	31 (40.8)
section *Aspergillus*	
*Fumigati*	10 (32.3)
*Nigri* *	12 (38.7)
*Terrei*	1 (3.2)
*Flavi*	5 (16.1)
Multiple **	3 (9.7)
Antifungal therapy	55 (72.4)
Type of antifungal agents	
Mold-active azole	10 (18.2)
Polyene	43 (78.2)
Echinocandins	2 (3.6)
Antifungal agent applied before PCR	45 (59.2)

Data are presented as *n* (%) unless otherwise noted. BAL, bronchoalveolar lavage; GM, galactomannan; IPA, invasive pulmonary aspergillosis; PCR, polymerase chain reaction. * Section *Nigri* included two isolates of *A. tubingensis*. ** Multiple included coinfection by *Aspergillus* section *Fumigati* and section *Nigri.*

**Table 3 jof-09-01192-t003:** Clinical performance of *benA* multiplex probe *Aspergillus* PCR and Aspergenius kit.

Target Kit	Probe	Positive	Negative	PPV	NPV	Sensitivity	Specificity
*BenA*	*Ascomycetes*	120/120	114/120	100	95	95.24	100
	*A. fumigatus*	120/120	112/120	100	93.33	93.75	100
	*A. niger*	118/120	117/120	98.33	97.5	97.52	98.32
	*A. terreus*	120/120	115/120	100	95.83	96	100
	*A. flavus*	116/120	115/120	96.67	95.83	95.87	96.64
	*A. tubingensis*	120/120	120/120	100	100	100	100
Aspergenius	*A. species*	120/120	116/120	100	96.67	96.77	100
	*A. fumigatus*	120/120	116/120	100	96.67	96.77	100
	*A. terreus*	120/120	120/120	100	100	100	100

*benA*, β-tubulin; A., *Aspergillus*; PPV, positive predictive value; NPV, negative predictive value.

**Table 4 jof-09-01192-t004:** Performance of *Aspergillus* PCR and sensitivity analysis.

**A. Baseline analysis: Possible IPA were included in non-IPA cases**				
**Variables**	**Estimate**	**Standard error**	**95% CI**	
Sensitivity	0.520	0.099	0.324	0.716
Specificity	0.647	0.066	0.516	0.778
PPV	0.419	0.089	0.246	0.593
NPV	0.733	0.066	0.604	0.863
LR+	1.473	0.397	0.694	2.253
LR−	0.741	0.172	0.403	1.079
DOR	1.988			
**B. Sensitivity analysis 1: Possible IPA were included in IPA cases**				
**Variables**	**Estimate**	**Standard error**	**95% CI**	
Sensitivity	0.454	0.067	0.323	0.586
Specificity	0.714	0.099	0.521	0.908
PPV	0.806	0.071	0.667	0.945
NPV	0.333	0.070	0.195	0.471
LR+	1.591	0.597	0.421	2.761
LR−	0.763	0.141	0.486	1.040
DOR	2.085			
**C. Sensitivity analysis 2: Possible IPA were excluded altogether**				
**Variables**	**Estimate**	**Standard error**	**95% CI**	
Sensitivity	0.520	0.099	0.324	0.716
Specificity	0.714	0.098	0.521	0.907
PPV	0.684	0.106	0.475	0.893
NPV	0.556	0.095	0.368	0.743
LR+	1.820	0.718	0.411	3.228
LR−	0.672	0.167	0.343	1.001
DOR	2.708			

CI, confidence interval; DOR, diagnostic odds ratio; IPA, invasive pulmonary aspergillosis; LR+, positive likelihood ratio; LR−, negative likelihood ratio; NPV, negative predictive value; PCR, polymerase chain reaction; PPV, positive predictive value.

## Data Availability

The data presented in this study are fully available on request from the corresponding author. The data are not publicly available because sequence and methods used in this study are patented.

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
