# Peer review of "Clinical Implementation of β-Tubulin Gene-Based Aspergillus Polymerase Chain Reaction for Enhanced Aspergillus Diagnosis in Patients with Hematologic Diseases: A Prospective Observational Study"

_jof, 2023, doi:10.3390/jof9121192_

Round 1
Reviewer 1 Report
Comments and Suggestions for Authors
The paper is well constructed and written, albeit with a significant number of stylistic errors according to established journal standards of academic English. The study is, however, justified appropriate with references appropriate to the body text. The results are well presented, with graphical figures appropriately labelled and reflective of the results detailed within the body text. The overall substance of the manuscript will be of interest, in my judgment, to the wider readership of the Journal provided typographical and linguistic errors are finessed.
Comments on the Quality of English LanguageThe paper contains a significant number of minor errors of academic English, including (but not limited to) issues of tense - i.e., "Invasive pulmonary aspergillosis (IPA) has been a major concern in patients with hematologic malignancies" (Line 44); punctuation - i.e., "The β-tubulin gene (BenA) is the most common reference gene for detecting Aspergillus species Moreover" (Line 65); and capitalisation - i.e., "Cytochrome P450 Sterol 14-alpha demethylase" (Line 74).
Proprietary names [e.g.., Roche Light-158 Cycler® 480 Instrument II (Roche)] should be stated in accordance with usual academic conventions and the Journal editorial style.
Author Response
Reply to reviewer’s comments : Manuscript ID [jof-2737911]
We would like to thank the editor and reviewers for the invaluable comments that helped a great deal in improving our original manuscript. Please find attached our revised paper and below our point-by-point responses to your suggestions and comments.
Reviewer #1 comments:
General comments: The paper is well constructed and written, albeit with a significant number of stylistic errors according to established journal standards of academic English. The study is, however, justified appropriate with references appropriate to the body text. The results are well presented, with graphical figures appropriately labelled and reflective of the results detailed within the body text. The overall substance of the manuscript will be of interest, in my judgment, to the wider readership of the Journal provided typographical and linguistic errors are finessed.
Comments 1: The paper contains a significant number of minor errors of academic English, including (but not limited to) issues of tense - i.e., "Invasive pulmonary aspergillosis (IPA) has been a major concern in patients with hematologic malignancies" (Line 44); punctuation - i.e., "The β-tubulin gene (BenA) is the most common reference gene for detecting Aspergillus species Moreover" (Line 65); and capitalisation - i.e., "Cytochrome P450 Sterol 14-alpha demethylase" (Line 74).
RESPONSE: Thank you for your valuable comments. We corrected errors in academic English and made sentences more flexible. The contents of Lines 44-46 were modified as follows. Invasive pulmonary aspergillosis (IPA) has been a significant concern among patients with hematologic malignancies and is emerging as a threat for individuals with severe respiratory viral infections, including coronavirus disease 2019 (COVID-19) and influenza. Similarly, Lines 69-67 was modified as follows: Although the β-tubulin gene (benA) may have lower sensitivity compared to commonly used ITS-based methods, it is effective for detailed identification of Aspergillus species and is used as a marker for phylogenetic analysis of filamentous Ascomycetes. All errors in academic terminology, including Line 74, have been corrected (e.g., gene benA in lowercase and italics, probe BenA in sperm and uppercase).
Comments 2: Proprietary names [e.g.., Roche Light-158 Cycler® 480 Instrument II (Roche)] should be stated in accordance with usual academic conventions and the Journal editorial style.
RESPONSE: Edited using Roche Light-158 Cycler® 480 Instrument II (Roche Diagnostics, Basel, Switzerland) and later written using Roche Co.
Reviewer 2 Report
Comments and Suggestions for Authors
In this manuscript, Lee at al describe the use of B-tubulin gene-based Aspergillus PCR of BAL fluid for Aspergillus diagnosis in patients with underlying hematologic diseases. This study strength is its prospective design, decent sample size cohort (76 BAL samples in 75 subjects) and interest to the reader, as improved fungal diagnostics are necessary.
The authors are clear about the limitations of their study and do not overstate their conclusions, particularly noting the reliable performance of this BenA PCR and its ability to detect a wide range of Aspergillus species.
However, the main concern I have is the performance of this assay compared to current standard of care/available assays. The authors' analyses of the performance of the PCR BAL would be better assessed next to the standard of care testing (galactomannan) that the authors confirm was collected in all BAL samples, as well as other commercially available PCR methods (like the Aspergenius) that the authors compared in terms of analytical performance of spiked samples, but then not in the prospective cohort. This would add to the investigators' study to see if some of the limitations they note (ie, impact of pre-testing antifungals, test specificity) may be also present with these already available tests. I think this would be particularly relevant given the poor sensitivity of the BenA assay demonstrated in the paper, which the authors only somewhat touch on in the Discussion.
Author Response
Reply to reviewer’s comments : Manuscript ID [jof-2737911]
We would like to thank the editor and reviewers for the invaluable comments that helped a great deal in improving our original manuscript. Please find attached our revised paper and below our point-by-point responses to your suggestions and comments.
Reviewer #2 comments:
General comments: In this manuscript, Lee at al describe the use of B-tubulin gene-based Aspergillus PCR of BAL fluid for Aspergillus diagnosis in patients with underlying hematologic diseases. This study strength is its prospective design, decent sample size cohort (76 BAL samples in 75 subjects) and interest to the reader, as improved fungal diagnostics are necessary.
The authors are clear about the limitations of their study and do not overstate their conclusions, particularly noting the reliable performance of this BenA PCR and its ability to detect a wide range of Aspergillus species.
Comments 1: However, the main concern I have is the performance of this assay compared to current standard of care/available assays. The authors' analyses of the performance of the PCR BAL would be better assessed next to the standard of care testing (galactomannan) that the authors confirm was collected in all BAL samples, as well as other commercially available PCR methods (like the Aspergenius) that the authors compared in terms of analytical performance of spiked samples, but then not in the prospective cohort. This would add to the investigators' study to see if some of the limitations they note (ie, impact of pre-testing antifungals, test specificity) may be also present with these already available tests. I think this would be particularly relevant given the poor sensitivity of the BenA assay demonstrated in the paper, which the authors only somewhat touch on in the Discussion.
RESPONSE: Thank you for your comment. We fully agree with the opinion regarding the limitations of not being able to conduct additional comparative evaluation in clinical samples between ITS-based and benA-based Aspergillus PCR. We were unable to perform a comparative evaluation for the entire clinical samples due to the shortage of specimen quantity and accessibility issues with ITS-based commercially available kits. We have described this as a limitation to enhance reader comprehension and underscore the imperative for future research. Please see lines 378-381.
Reviewer 3 Report
Comments and Suggestions for Authors
Some suggestions for improving the manuscript.
Expand a little more in the introductory section on the organism in question. Why its detection is important, some clinical data motivate the search for rapid and safe detection of the pathogen.
Explain a little more about the molecular markers (how many are needed) that are needed to use them as references for detecting the fungus.
In the supplementary documents; submit the approval document by the ethics committee to carry out said research.
line 120: Full name of organisms not mentioned yet.
line 122: ml --> mL
Fig. 1. ml --> mL; ul --> uL
Correct some errors when writing the names of some organisms.
lines 299-300: Perhaps the work will increase the impact if you expand the information on this aspect and the possibility of detecting those that are mutants and that this mutation confers resistance to antifungals.
Fig.2. Place the name of the DNA marked as references (legend)
Author Response
Reply to reviewer’s comments : Manuscript ID [jof-2737911]
We would like to thank the editor and reviewers for the invaluable comments that helped a great deal in improving our original manuscript. Please find attached our revised paper and below our point-by-point responses to your suggestions and comments.
Reviewer #3 comments:
Comments 1: Expand a little more in the introductory section on the organism in question. Why its detection is important, some clinical data motivate the search for rapid and safe detection of the pathogen.
RESPONSE: We appreciate your comment. We totally agree with the opinions regarding the clinical significance of invasive aspergillosis and the importance of early diagnosis. I have added relevant content to the introduction section to refresh the readers' understanding of the importance of rapid testing for the PCR diagnosis and resistance of invasive aspergillosis (IA). This aims to provide clarity on the purpose of our study. Please see lines 48-50.
Comments 2: Explain a little more about the molecular markers (how many are needed) that are needed to use them as references for detecting the fungus.
RESPONSE: In lines 255-258, we specified the use of Aspergillus positive control (APC) as reference DNA, employing a serial dilution of the gene copy number in the experiment. The copy numbers, ranging from 100 to 105, are detailed in both the text and the corresponding figure.
Comments 3: In the supplementary documents; submit the approval document by the ethics committee to carry out said research.
RESPONSE: Thank you for your comment. The institutional review board approval letter for this study is attached as a supplementary document.
Comments 4: line 120: Full name of organisms not mentioned yet.
line 122: ml --> mL
Fig. 1. ml --> mL; ul --> uL
Correct some errors when writing the names of some organisms.
RESPONSE: Thank you for your comments. The mentioned section has been revised throughout the entire text. Modifications were made using special characters, such as ㎖ and ㎕, and the names of organisms were also adjusted to comply with international standards.
Comments 5: lines 299-300: Perhaps the work will increase the impact if you expand the information on this aspect and the possibility of detecting those that are mutants and that this mutation confers resistance to antifungals.
RESPONSE: We revised the notation for mutations in Figure 3 to TR34 for improved readability and accurate communication. Please see Figure 3.
Comments 6: Fig.2. Place the name of the DNA marked as references (legend)
RESPONSE: We have included the name of the DNA marker used as a reference in the figure legend. As mentioned in the text, it is the Aspergillus positive control (APC) designed and prepared in a previous in vitro experiment. The reference used is number 19. Please see Figure 2.
Reviewer 4 Report
Comments and Suggestions for Authors
Dear authors,
Your paper entitled: Clinical implementation of β-tubulin gene-based Aspergillus PCR for
enhanced Aspergillus diagnosis in patients with hematologic diseases: A
prospective observational study, is very nice written article; is a presentation of attempts for design and possible establishing of molecular methods for rapid and accurate diagnosis of pulmonary invasive aspergillosis, infection with very high morbidity and mortality.
I recommend this paper for publication after minor revision
This study concerns important infections. Moreover, in the opinion of many clinical microbiologists, as well as clinicians, invasive fungal infections are neglected and many commercial rapid non-culture tests are not available to many laboratories. There are many attempts in the design, validation and standardization of molecular analyzes that would provide fast and accurate diagnosis, that would not take long, and that would be easy to process, and even more important, in interpretation. In parallel, research is still being conducted with the aim of developing diagnostic tests for the fast determination of resistant species, causative agents.
Additionally, you presented molecular analyses for detection of genes that determined resistant species of genus Aspergillus, which could provide appropriate treatment. The survival of high-risk patients with IPA primarily depends on timely and accurate diagnosis and established antifungal therapy. Aware of the fact that Aspergillus resistance to azoles is increasing, presentation of new methods is of great significance.
However, my general suggestions are:
It is very important to emphasize that despite new methods in diagnostics, a multidisciplinary approach is always necessary (since that the main drawback and limitation in diagnostics is the inability to distinguish colonization from invasive disease, as well as possibility of high risk of contamination of samples with spores from the air = false positive results)
It will be better to be explained definition of proven, probable and possible invasive pulmonary aspergillosis, since that new molecular analyses change the redistribution of IPA.
Listing Aspergillus sections would be better replaced by complexes
Pay attention that all abbreviations in figures and tables are explained in the legends
Comments on the Quality of English LanguageModerate editing of English language required
Author Response
Reply to reviewer’s comments : Manuscript ID [jof-2737911]
We would like to thank the editor and reviewers for the invaluable comments that helped a great deal in improving our original manuscript. Please find attached our revised paper and below our point-by-point responses to your suggestions and comments.
Reviewer #4 comments:
General comments: Your paper entitled: Clinical implementation of β-tubulin gene-based Aspergillus PCR for
enhanced Aspergillus diagnosis in patients with hematologic diseases: A prospective observational study, is very nice written article; is a presentation of attempts for design and possible establishing of molecular methods for rapid and accurate diagnosis of pulmonary invasive aspergillosis, infection with very high morbidity and mortality.
I recommend this paper for publication after minor revision
This study concerns important infections. Moreover, in the opinion of many clinical microbiologists, as well as clinicians, invasive fungal infections are neglected and many commercial rapid non-culture tests are not available to many laboratories. There are many attempts in the design, validation and standardization of molecular analyzes that would provide fast and accurate diagnosis, that would not take long, and that would be easy to process, and even more important, in interpretation. In parallel, research is still being conducted with the aim of developing diagnostic tests for the fast determination of resistant species, causative agents.
Additionally, you presented molecular analyses for detection of genes that determined resistant species of genus Aspergillus, which could provide appropriate treatment. The survival of high-risk patients with IPA primarily depends on timely and accurate diagnosis and established antifungal therapy. Aware of the fact that Aspergillus resistance to azoles is increasing, presentation of new methods is of great significance.
However, my general suggestions are:
Comments 1: It is very important to emphasize that despite new methods in diagnostics, a multidisciplinary approach is always necessary (since that the main drawback and limitation in diagnostics is the inability to distinguish colonization from invasive disease, as well as possibility of high risk of contamination of samples with spores from the air = false positive results)
RESPONSE: We totally agree with your comments. When we introduced new diagnostic method, it is important to use multimodal approaches. As articulated in the discussion section, we advocate for the combined use of PCR with GM to amplify sensitivity and facilitate the discrimination of contamination, as detailed in lines 333-336.
Comments 2: It will be better to be explained definition of proven, probable and possible invasive pulmonary aspergillosis, since that new molecular analyses change the redistribution of IPA.
RESPONSE: Thank you for your invaluable comment. I agree with the importance of providing an explanation for the use of the EORTC/MSG criteria-based classification for the diagnosis of invasive pulmonary aspergillosis (IPA) throughout this study. Related information was described in the definition section of the Method and additional information has been added to facilitate a clearer understanding for our readers. Please see lines 109-111.
Comments 3: Listing Aspergillus sections would be better replaced by complexes
RESPONSE: Thank you for your valuable comment. We acknowledge and agree with your suggestion to consider using complexes instead of sections in certain areas. However, we hope for your understanding as the decision to use sections was a collective one made by all authors. In this study, our focus was on evaluating the analytic performance of benA-based PCR for discriminating species within each Aspergillus section. The main objective and strength of our research lie in the precise detection of species within these sections, making the term "sections" more fitting than "complexes." Furthermore, our team has previously employed sections in benA-based PCR research, and we have maintained consistency in using this terminology throughout our manuscript. We appreciate your consideration of these points.
Comments 4: Pay attention that all abbreviations in figures and tables are explained in the legends
RESPONSE: Thank you for your comments. We have conducted a comprehensive review of all tables and figures and revised legends and figures. Please see Figure 1-3.
Reviewer 5 Report
Comments and Suggestions for Authors
The study delves into assessing the efficacy of a particular diagnostic approach while exploring various methodologies and definitions. The introduction provides a robust foundation by delineating the challenges inherent in diagnosing Invasive Pulmonary Aspergillosis (IPA). It meticulously outlines the shortcomings of current diagnostic techniques, such as the limitations of galactomannan (GM) testing and PCR-based methods due to issues like accuracy, susceptibility to interference from antifungal treatments, and cost implications. Notably, it underscores the potential of the BenA gene-based PCR assay in mitigating these limitations by enhancing accuracy and sensitivity, particularly in discerning between different Aspergillus species.
Regarding enhancements, I would suggest augmenting the introduction by expounding on the distinct characteristics of the Aspergillus genus. This would include an in-depth discussion about the various species within each section (Nigri, Fumigati, etc.), along with epidemiological data encompassing global prevalence, mortality rates, morbidity statistics, available treatment options, associated costs, and the emergence of antifungal resistance.
Furthermore, in the discussion section, I would recommend a more extensive comparative analysis between the newly described method and previously utilized approaches. Specifically, there could be a comprehensive exploration of the time and financial implications associated with adopting this new diagnostic technique compared to established methods.
Lastly, to ensure precision, it's advisable to italicize all Latin names (e.g., lines 231, and 233) throughout the manuscript for consistency and accuracy.
Author Response
Reply to reviewer’s comments : Manuscript ID [jof-2737911]
We would like to thank the editor and reviewers for the invaluable comments that helped a great deal in improving our original manuscript. Please find attached our revised paper and below our point-by-point responses to your suggestions and comments.
Reviewer #5 comments:
General comments: The study delves into assessing the efficacy of a particular diagnostic approach while exploring various methodologies and definitions. The introduction provides a robust foundation by delineating the challenges inherent in diagnosing Invasive Pulmonary Aspergillosis (IPA). It meticulously outlines the shortcomings of current diagnostic techniques, such as the limitations of galactomannan (GM) testing and PCR-based methods due to issues like accuracy, susceptibility to interference from antifungal treatments, and cost implications. Notably, it underscores the potential of the BenA gene-based PCR assay in mitigating these limitations by enhancing accuracy and sensitivity, particularly in discerning between different Aspergillus species.
Comments 1: Regarding enhancements, I would suggest augmenting the introduction by expounding on the distinct characteristics of the Aspergillus genus. This would include an in-depth discussion about the various species within each section (Nigri, Fumigati, etc.), along with epidemiological data encompassing global prevalence, mortality rates, morbidity statistics, available treatment options, associated costs, and the emergence of antifungal resistance.
RESPONSE: Thank you for your comment. We agree with the opinions regarding the clinical significance of invasive aspergillosis and the importance of introducing characteristics of Aspergillus. I have added relevant content to the introduction section to refresh the readers' understanding of Aspergillus. Please see lines 48-50 and 75-76. This research, however, aims to validate a benA-based PCR method for the early diagnosis and resistance detection of Aspergillus, and to discuss its clinical application. Consequently, we kindly request your understanding that we refrained from providing an exhaustive detail of Aspergillus species in the introduction, aligning with the authors' perspective that an overly detailed description could disrupt the paper's consistent flow and diminish reader concentration.
Comments 2: Furthermore, in the discussion section, I would recommend a more extensive comparative analysis between the newly described method and previously utilized approaches. Specifically, there could be a comprehensive exploration of the time and financial implications associated with adopting this new diagnostic technique compared to established methods.
RESPONSE: Thank you, reviewer, for your favorable assessment. In response, we have incorporated the following details into the discussion to further emphasize the strengths of our study. Please see lines 345-354.
Comments 3: Lastly, to ensure precision, it's advisable to italicize all Latin names (e.g., lines 231, and 233) throughout the manuscript for consistency and accuracy.
RESPONSE: As you mentioned, we have revised it to comply with international paper standards and MDPI standards.